# Solid-Phase Synthesis of the Bicyclic Peptide OL-CTOP Containing Two Disulfide Bridges, and an Assessment of Its In Vivo μ-Opioid Receptor Antagonism after Nasal Administration

**DOI:** 10.3390/molecules28041822

**Published:** 2023-02-15

**Authors:** Ramanjaneyulu Rayala, Annika Tiller, Shahayra A. Majumder, Heather M. Stacy, Shainnel O. Eans, Aleksandra Nedovic, Jay P. McLaughlin, Predrag Cudic

**Affiliations:** 1Department of Chemistry and Biochemistry, Charles E. Schmidt College of Science, Florida Atlantic University, 777 Glades Road, Boca Raton, FL 33431, USA; 2Department of Pharmacodynamics, School of Pharmacy, University of Florida, 1345 Center Drive, Gainesville, FL 32610, USA

**Keywords:** intranasal delivery, odorranalectin, CTOP, disulfide bond, solid-phase synthesis, trypsin, μ-opioid receptor, morphine, antagonism, respiratory depression

## Abstract

New strategies facilitate the design of cyclic peptides which can penetrate the brain. We have designed a bicyclic peptide, OL-CTOP, composed of the sequences of a selective μ-opioid receptor antagonist, CTOP (f-*cyclo*(CYwOTX)T) (X = penicillamine, Pen; O = ornithine) and odorranalectin, OL (YASPK-*cyclo*(CFRYPNGVLAC)T), optimized its solid-phase synthesis and demonstrated its ability for nose-to-brain delivery and in vivo activity. The differences in reactivity of Cys and Pen thiol groups protected with trityl and/or acetamidomethyl protecting groups toward I_2_ in different solvents were exploited for selective disulfide bond formation on the solid phase. Both the single step and the sequential strategy applied to macrocyclization reactions generated the desired OL-CTOP, with the sequential strategy yielding a large quantity and better purity of crude OL-CTOP. Importantly, intranasally (i.n.s.) administered OL-CTOP dose-dependently antagonized the analgesic effect of morphine administered to mice through the intracerebroventricular route and prevented morphine-induced respiratory depression. In summary, the results demonstrate the feasibility of our solid-phase synthetic strategy for the preparation of the OL-CTOP bicyclic peptide containing two disulfide bonds and reveal the potential of odorranalectin for further modifications and the targeted delivery to the brain.

## 1. Introduction

Opioid medication misuse, addiction, and overdose are growing public health problems nationwide [1,2,3]. Methadone, buprenorphine, naltrexone and naloxone are the only FDA-approved medications to treat opioid use disorder (OUD) and, in the case of opioid receptor antagonists naltrexone and naloxone, overdose. However, these medications possess clinical shortcomings, prompting an urgent need for novel and more efficient medications to treat OUD and to effectively counteract overdoses of very potent opioids such as fentanyl [4,5,6]. Given their potency, peptides represent particularly attractive leads for the development of novel drugs that target the brain. For instance, H-D-Phe-*cyclo*(Cys-Tyr-D-Trp-Orn-Thr-Pen)-Thr-NH_2_ (CTOP, where Pen = penicillamine and Orn = ornithine) [7,8,9] is an antagonist with high affinity (2.8 nM) and selectivity for μ-opioid receptors (MOR). CTOP-mediated MOR antagonism is reportedly 10–400 times more potent than naloxone [9], suggesting it might better antagonize high-potency MOR agonists such as fentanyl. However, as a peptide, CTOP does not penetrate the brain after systemic administration. Both the blood-brain barrier (BBB) and blood-cerebrospinal fluid barrier (BCB) restrict the transport of therapeutic peptides like CTOP from the systemic circulation into the CNS [10,11]. Intranasal (i.n.s.) administration may enable peptide therapeutics to directly enter the brain by bypassing the BBB and BCB [11,12]. This delivery route exploits the olfactory or trigeminal cranial nerve systems, which initiate in the brain and terminate in the nasal cavity at the olfactory neuroepithelium or respiratory epithelium [11,12]. Once the drugs are delivered from the submucosal space of the nose into the cerebrospinal fluid (CSF) compartment of the brain, drugs are able to disperse throughout the brain. Intranasal delivery avoids systemic circulation of the drug and reduces the risk of systemic side effects as well as hepatic/renal clearance, elongating half-life (t_1/2_) and efficacy of the drug administered. For example, oxytocin (Pitocin) has a plasma t_1/2_ of about 1–6 min, whereas in the extracellular space of the brain and in the CSF, this time increases to approximately 30 min. [13,14]. Similar increases in t_1/2_ have been reported for other peptides, including β-endorphin [15,16,17]. These studies demonstrated that CNS concentrations of peptides are higher over a longer period of time, indicating that the i.n.s. administration of peptides may be considerably more effective compared to other administration routes.

Presently, to address the need for both improved medications to treat OUD and overdose, we designed and synthesized a unique and novel bicyclic peptide composed of the sequences of the potent and selective MOR antagonist, CTOP [7,8,9] and odorranalectin (OL; YASPK-*cyclo*(CFRYPNGVLAC)T), a naturally occurring carbohydrate-binding cyclic peptide [18]. Exhibiting lectin-like properties, odorranalectin preferentially binds to L-fucose (Fuc) and, to a lesser extent, to D-galactose (Gal) and *N*-acetyl-D-galactosamine (GalNAc) [18,19,20], all of which are widely distributed on the olfactory epithelium of nasal mucosa [21]. The residues critical for odorranalectin-glycoprotein binding have been mapped through alanine scanning, further demonstrating the key interactions [20]. This suggests the feasibility of odorranalectin and OL-CTOP interacting with these sugar moieties, hence leading to the extension of the residence time in the nasal cavity by these peptides and increased adsorption into the brain by bypassing the BBB and BCB [11,12]. We theorized that incorporation of the CTOP sequence into the odorranalectin scaffold would thus facilitate potent MOR antagonism directly within the brain. As agonist activation of MOR within respiratory centers in the medulla and brainstem is known to mediate opioid respiratory depression [22], the targeted delivery of a potent antagonist to MOR in the brain would be a potentially valuable tool to reverse opioid respiratory depression and overdose [23,24,25,26,27].

Herein, we describe a solid-phase synthesis of bicyclic peptide OL-CTOP containing two disulfide bridges, its stability toward proteolytic hydrolysis and its initial in vivo characterization in mice for ΜOR-selective antagonist properties and the ability to protect against morphine-induced respiratory depression.

## 2. Results

### 2.1. Synthesis

To establish the optimal conditions for the formation of the disulfide bridges during the solid-phase synthesis of the bicyclic OL-CTOP peptide, various I_2_ concentrations, oxidation times, and solvents CH_2_Cl_2_ or DMF with or without the addition of DMSO were examined. We also exploited the differences in oxidation reaction rates of cysteine (Cys) and penicillamine (Pen) side-chain thiols protected with trityl (S-Trt) and acetamidomethyl (S-Acm) in different solvents such as CH_2_Cl_2_ or DMF for the formation of proper disulfide bridges. Thus, Cys(Acm), Cys(Trt) and Pen(Trt) were strategically positioned within the sequence of the linear peptidyl-resin precursor to allow the formation of a bicyclic peptide with a CTOP cyclic sequence grafted into the odorranalectin scaffold. Standard SPPS Fmoc-chemistry methodology was used throughout the synthesis of peptidyl-resin precursor (Figure 1).

Our synthetic strategy included the oxidation of Cys^10^ and Pen^15^, protected with Trt in CH_2_Cl_2_, followed by the oxidation of Cys^6^ and Cys^17^, protected with Acm in DMF. To trap the carbocations produced during the reaction and to provide some protection to Tyr and Trp residues against alkylation, both present in the sequence of the OL-CTOP peptide, anisole was used as a scavenger in all oxidation reactions [28]. All reactions were carried out at room temperature. The formation of the cyclic and bicyclic peptide products was monitored by RP-HPLC and MALDI-ToF MS following the mini cleavage of the peptide from the resin. The optimal conditions for Cys(Trt)^10^ and Pen(Trt)^15^ oxidation included 2 molar eq of I_2_, a reaction time of 20 min, and CH_2_Cl_2_ as a solvent. Under these reaction conditions, approximately 67% of the monocyclic product **1** with Cys(Acm)_6_ and Cys(Acm)^17^ and 25% of the monocyclic peptides **2** and **3** with partially removed Acm protecting group from Cys^6^ or Cys^17^ were isolated (Figure 1A). No linear peptide precursor was detected under the applied analytical conditions. The addition of DMSO as a co-solvent to the reaction mixture did not improve the S-Trt oxidation yield. The addition of DMSO (30–150 molar eq) to the reaction mixture led to a slight reduction in the yield of the oxidation product, affording approximately 50% of the monocyclic product (Appendix A). As in the previous case, 30% of the cyclic peptides **2** and **3** with partially deprotected Cys^6/17^ were detected. In the next step, oxidation of a monocyclic peptidyl-resin precursor containing Cys(Acm)^6^ and Cys(Acm)^17^, as well as deprotected Cys^6^ or Cys^17^, was carried out in DMF. We found that 1.5 molar eq of I_2_ and 30 min reaction time gave the highest yield of the bicyclic product (approximately 85%) without detectable starting monocyclic precursor (Figure 1B). As in the case of S-Trt oxidation, DMSO reduced the oxidation yield of S-Acm, resulting in approximately 42% of the bicyclic product and 28% of a monocyclic precursor.

In all cases, the extension of the reaction time to more than 30 min significantly decreased the amount of the bicyclic peptide products and respective precursors, whereas the number of byproducts increased. To further take advantage of different oxidation reaction rates of S-Acm and S-Trt with I_2_ in different solvents, we explored the possibility of the solid-phase synthesis of bicyclic OL-CTOP peptide in one step (Figure 1). DMF was used as a solvent because the difference in oxidation rates of S-Trt and S-Acm is not large, as is the case for CH_2_Cl_2_, allowing for a reduction of the overall oxidation time and reduction of the number and quantity of side-products. Based on the optimized sequential solid-phase synthesis of OL-CTOP, 4 eq of I_2_ and a 30 min reaction time were selected. Under these conditions, 24% of the bicyclic OL-CTOP product and 65% of the monocyclic peptide precursor with partially deprotected Cys^6/17^ were detected (Figure 1C). However, the oxidation reaction had to be repeated an additional three times to completely consume the monocyclic precursor. Under these conditions, approximately 66% of the bicyclic OL-CTOP product was obtained, with the remaining side products representing impurities that were not identified. In all cases, after HPLC-purification of the crude peptide, approximately 35–40% of the pure bicyclic product (>90% purity) was obtained. All synthesized peptides were characterized by MALDI-ToF MS and RP-HPLC (see Appendix A).

### 2.2. Proteolytic Stability

To investigate the stability of OL-CTOP toward proteolytic degradation, we incubated OL-CTOP with trypsin immobilized on agarose resin (as it simplifies sample separation after the reaction) and monitored its proteolysis. Analysis of OL-CTOP degradation by RP HPLC and MALDI-TOF mass spectrometry revealed that the *N*-terminal part of the OL-CTOP sequence, YASPK, was hydrolyzed within the first 10 min, without a detectable trace of the parent peptide (OL-CTOP: R_t_ = 16.71 min, found [M^+^H]^+^
*m*/*z* = 2216.99; calculated *m*/*z* = 2217.96, hydrolysis product: R_t_ = 17.26 min, found [M^+^H]^+^
*m*/*z* = 1669.67, calculated *m*/*z* = 1671.01) (see Appendix A). After 1 h of OL-CTOP incubation with trypsin, the peptide bond between Arg^8^ and phe^9^ was also cleaved, leading to the additional cyclic fragment f-*cyclo*(CYwOTX)TC(CFR) (R_t_ = 15.88 min, [M^+^H]^+^ found *m*/*z* = 1689.0, calculated *m*/*z* = 1687.68, Figure 2B). Importantly, this trypsin degradation fragment has the active CTOP sequence, and its quantity increases over the 24 h time period to become the major degradation product (Figure 2C).

### 2.3. Carbohydrate-Binding Study Using Isothermal Titration Calorimetry (ITC)

For efficient nose-to-brain transport of OL-CTOP, it is highly desirable that this peptide exhibits affinity toward carbohydrates expressed on the olfactory epithelium of nasal mucosa. We have shown previously that odorranalectin binds to model glycoproteins containing terminal Fuc, Gal and GalNAc sugar moieties with μM affinities and identified the binding interface [19,20]. Based on these studies, we modified the odorranalectin scaffold by grafting in the CTOP sequence and hypothesized that this novel bicyclic peptide, OL-CTOP, would exhibit the functional properties of both parent peptides; carbohydrate binding of odorranalectin and opioid antagonist activity of CTOP. To demonstrate that OL-CTOP is capable of binding relevant carbohydrates, we conducted a binding study using asialofetuin (ASF) as a model system. ASF was chosen because it is a well-characterized glycoprotein that possesses three triantennary *N*-linked oligosaccharides with terminal Gal residues and three *O*-linked disaccharide Galβ1-3GalNAc chains available for binding to odorranalectin and its OL-CTOP derivate. The binding affinities of OL-CTOP toward ASF were determined by Isothermal Titration Calorimetry (ITC). As shown in Figure 3, OL-CTOP binds ASF with *K*_d_ of 176 μM. The obtained *K*_d_ is comparable to our reported value for odorralnalectin/ASF binding [19,20]. The enthalpy (ΔH) of the binding interaction between OL-CTOP and ASF was −131 kJ/mol, whereas the entropy (−TΔS) was 109.5 kJ/mol, showing that the OL-CTOP/ASF interaction is an enthalpy-driven process, which is typical for interactions between lectins and carbohydrate ligands [29,30]. This suggests the possibility for binding of OL-CTOP to carbohydrates expressed on the olfactory epithelium that may lead to the extended residence time of OL-CTOP in the nasal cavity and thereby increased adsorption into the brain.

### 2.4. In Vivo Pharmacological Evaluation

MOR antagonism by CTOP was confirmed in vivo by pretreating mice with the parent peptide and then administering intracerebroventricular (i.c.v.) morphine (10 nmol) prior to testing tail-withdrawal latency in the 55 °C warm-water tail-withdrawal test. When a dose validated to produce MOR antagonism was administered directly to the brain through the intracerebroventricular route, CTOP (637 μg, or 3 nmol) significantly antagonized the effect of morphine (F_(2,21)_ = 60.4, *p* < 0.0001; one-way ANOVA with Tukey post hoc test; Figure 4A). In contrast, when administered intranasally (i.n.s.), the parent peptide CTOP (600 μg) was unable to antagonize the effects of morphine (*p* = 0.99, Tukey post hoc test; Figure 4A).

The ability of intranasally administered OL-CTOP to antagonize the analgesic effect of morphine in vivo was then assessed in mice using the 55 °C warm-water tail-withdrawal test (Figure 4). Graded doses of OL-CTOP (100, 300 or 600 μg) were first administered through the intranasal (i.n.s.) route. OL-CTOP alone produced small (~1 s) but significant increases in tail withdrawal latency from baseline responses out to 30 min post-administration (time x treatment, *F*_(6,48)_ = 3.04, *p* = 0.01; two-way RM ANOVA; Figure 4B, left panel).

Following the confirmed return to baseline values at 45 min, all OL-CTOP-treated mice, including a group treated i.n.s. with 30 μg, were administered intracerebroventricular (i.c.v.) morphine (10 nmol), and the tail-withdrawal latency assessed 20, 30 and 40 min afterward, (Figure 4B, right panel). Control (naïve) mice administered morphine showed a significant increase in tail-withdrawal latency over the baseline value of 1.49 ± 0.09 s (*F*_(3,28)_ = 63.6, *p* < 0.0001; one-way ANOVA; Figure 4B, blue circles in the right panel). In contrast, mice treated intranasally with OL-CTOP demonstrated a significant general reduction in the antinociceptive effect of morphine across doses tested (time × treatment, *F*_(8,58)_ = 3.39, *p* = 0.003; two-way RM ANOVA; Figure 4B, right panel). Notably, the magnitude of OL-CTOP-induced antagonism of morphine was not significantly different across pretreatment doses (*p* = 0.88 to 0.99 between responses, Tukey’s post hoc testing).

#### 2.4.1. Opioid Receptor Selectivity of OL-CTOP Antagonist Activity

OL-CTOP was evaluated for antagonist selectivity against the MOR-preferring agonist morphine (10 nmol, i.c.v.), the KOR-selective agonist U50,488 (100 nmol, i.c.v.), and the DOR-selective agonist SNC-80 (100 nmol, i.c.v.; Figure 5). A 100 μg, i.n.s. pretreatment with OL-CTOP exhibited significant antagonism (interaction; *F*_(2,45)_ = 5.78, *p* = 0.006; two-way ANOVA) that was selective for morphine (*p* = 0.003; Sidak’s post hoc test; Figure 4), but not U50,488 (*p* = 0.99) or SNC-80 (*p* = 0.67).

#### 2.4.2. Evaluation of Intranasal OL-CTOP Protection from Morphine-Induced Respiratory Depression

Mice were administered intranasal saline or OL-CTOP (100 or 600 μg) 45 min prior to intraperitoneal morphine (20 mg/kg). As expected, the control saline-treated mice given morphine demonstrated significant, time-dependent respiratory depression compared to mice treated with vehicle alone (10–60 min; *F*_(15,290)_ = 14.3, *p* < 0.0001, two-way RM ANOVA with Tukey’s multiple comparison *post hoc* test; Figure 6). OL-CTOP significantly prevented morphine-induced respiratory depression after pretreatment with 100 μg († *p* = 0.05 at 20–40 min and *p* = 0.02 at 40–60 min; Tukey’s test, Figure 6) or 600 μg († *p* ≤ 0.009, 0–60 min; Tukey’s test, Figure 6), consistent with a dose-dependent MOR antagonism. Notably, although mice receiving a 100 μg OL-CTOP pretreatment before morphine showed no significant differences in respiration rate as compared to vehicle-treated mice (*p* = 0.06, 0.55 and 0.75 at points out to 60 min; Figure 6), they did show elevated respiration afterward (* *p* ≤ 0.03, 60–120 min; Figure 6), an effect seen even earlier with mice pretreated with the 600 μg dose of OL-CTOP (* *p* ≤ 0.04, 20–120 min; Figure 6).

## 3. Discussion

With the growing interest in peptides as lead structures for drug discovery, a continuing need exists for synthetic modifications providing more stable and bioactive analogs [31,32,33]. Among the approaches to improve peptide pharmaceutical properties, the cyclization of linear peptides is an attractive method to provide these analogs [34,35,36,37]. Peptide cyclization introduces conformational constraints to a polypeptide backbone, significantly decreasing the conformational freedom that may lead to preorganized conformations better suited for binding a target protein [38,39]. The rigidity of cyclic peptides minimizes the entropic penalty upon binding, allowing high affinity and selectivity [39,40,41]. In addition, cyclization has been shown to dramatically improve the stability of peptides against proteolytic degradation [42,43]. Structural rigidity and lack of both amino and carboxyl termini in cyclic peptides contribute to peptide resistance to hydrolysis by endo- and exo-peptidases present in blood [44]. In proteins, conformational restrictors are intramolecular disulfide bonds, reversible covalent bonds formed by the oxidation of the thiol groups of Cys residues. A disulfide bond may contribute up to 25 kJ/mol to the overall conformational stability of proteins at optimal temperatures [45]. For example, disulfide-rich venom peptides from snakes, scorpions or cone snails exhibit exceptional pharmacological properties and diverse biological activities. Thus, disulfide-rich peptides and short proteins have long been attractive targets for drug discovery [37,45,46,47,48,49,50]. Some peptides of this class include ziconotide, eptifibatide, and linaclotide, all already used as therapeutics [51,52,53]. Synthesis of these peptides and their analogs is particularly challenging, considering the presence of multiple disulfide bonds and the requirement for a specific peptide folding through proper Cys residue oxidation. Traditionally, there are two approaches to forming multiple intramolecular disulfide bonds in synthetic peptides; the single step and the sequential approaches [54,55]. In the single-step approach, all disulfide bonds are formed in one step by direct oxidation with air, DMSO or I_2_. The success of this approach depends on the peptide’s ability to adopt the native conformation in the reaction buffer. On the other hand, the sequential approach requires the use of orthogonal protecting groups for Cys side-chain thiols [54,56]. In this case, each disulfide bond is formed sequentially from pairs of Cys residues bearing compatible thiol-protecting groups. Both strategies are applicable to solution or solid-phase peptide cyclization. Typically, the reaction is carried out under high-dilution conditions (or pseudo-dilution conditions for solid-phase cyclization) using standard oxidizing agents [57]. Considering potential difficulties with the solubility of peptides bearing protected Cys residues in the reaction buffer and requirements for purification after each step, peptide cyclization via Cys oxidation on a solid support is particularly attractive.

We sought to apply a solid-phase strategy to synthesize the odorranalectin analog containing the CTOP μ-opioid receptor antagonist, OL-CTOP. The OL-CTOP is designed based on previous studies by us [19,20] and others [18] showing that Lys^5^, Phe^7^, Tyr^9^, Gly^12^, Leu^14^, and Thr^17^ were important for odorranalectin binding to Fuc and, to a lesser extent, to Gal and GalNAc, monosaccharides expressed on the olfactory nerves. However, Lys^5^, Cys^6^, Phe^7^, Cys^16^ and Thr^17^ were critical for fucose binding [18,20]. Thus, we hypothesize that replacement of the odorranalectin amino acid sequence that is not directly involved in or it is less relevant for binding to olfactory monosaccharides with CTOP may provide a novel and unique analog with desired opioid activity while preserving carbohydrate affinity for successful nose-to-brain delivery. The ITC binding study shown in Figure 3 confirmed our hypothesis and demonstrated that OL-CTOP is capable of binding ASF, a model glycoprotein, with *K*_d_ = 176 μM. To establish the optimal conditions for the formation of the disulfide bridges during the solid-phase synthesis of bicyclic OL-CTOP peptide, various I_2_ concentrations, oxidation times, and solvents CH_2_Cl_2_ or DMF with or without the addition of DMSO were examined DMSO had previously been shown as a desirable co-solvent for disulfide bond formation due to its denaturing and catalytic properties during oxidations of Cys by I_2_ [58]. In addition, the reactivity of the Cys/Pen side-chain thiol groups protected with Trt and Acm toward I_2_ in various solvents is significantly different, allowing selective oxidation without the deprotection of the thiol groups [55,59].

For the solid-phase formation of disulfide bridges present in the OL-CTOP sequence, the different reactivity of Cys and Pen thiol groups protected with Trt and/or Acm protecting groups toward I_2_ in CH_2_Cl_2_ and DMF was exploited. This synthetic strategy began with the oxidation of Cys^10^ and Pen^15^ protected with Trt in CH_2_Cl_2_, followed by the oxidation of Cys^6^ and Cys^17^ protected with Acm in DMF, Figure 1. This approach was chosen based on the studies of Kamber and co-workers demonstrating remarkable selectivity (~1000-fold) between S-Trt and S-Acm oxidation with I_2_ in different solvents [59]. To trap the carbocations produced during the reaction and to provide some protection to Tyr and Trp residues against alkylation, both present in the sequence of the OL-CTOP peptide, anisole was used as a scavenger in all oxidation reactions as demonstrated effective in previous studies [28]. Under these conditions, approximately 67% of the monocyclic product **1** with Cys(Acm)^6^ and Cys(Acm)^17^ and 25% of the monocyclic peptides **2** and **3** with partially removed Acm protecting group from Cys^6^ or Cys^17^ were isolated, Figure 1. The observed Acm removal is not surprising, considering that I_2_ is commonly used for the deprotection of Cys(Acm). It is worth noting that no linear peptide precursor was detected under the applied analytical conditions, and the addition of DMSO to the reaction mixture did not improve the S-Trt oxidation yield.

In all cases, the extension of the reaction time to more than 30 min significantly decreased the yield of the bicyclic peptide products and respective precursors. The observed byproducts were a complex mixture, possibly composed of peptides with modified amino acid residues and over-oxidized Cys/Pen, as previously reported [60,61]. The effect of DMSO on the Cys/Pen thiol groups oxidation yields using I_2_ as an oxidant could possibly be explained by the formation of a charge-transfer complex between iodine and DMSO, which undergoes chemical transformations to HI and the corresponding sulfones and, thus, an unfavorable increase in acidity of the reaction mixture [62]. To further take advantage of different oxidation reaction rates of S-Acm and S-Trt with I_2_ in different solvents, we explored the possibility of the solid-phase synthesis of bicyclic OL-CTOP peptide in one step, Figure 1. As a solvent, we chose DMF over DCM because reaction rates for I_2_ oxidation of S-Trt and S-Acm are significantly shorter in DMF than in DCM [59]. In this way, we attempted to reduce the overall oxidation time and minimize the possibility of the formation of a large number and quantity of side products. Although this approach was successful in obtaining OL-CTOP, the oxidation reaction with I_2_ had to be repeated several times to completely consume the linear resin-peptidyl precursor, which in turn led to a significant increase of side products and lower yields of OL-CTOP, Figure 1C.

The development of therapeutic peptides also represents a challenge due to their susceptibility toward proteolytic degradation [63]. Trypsin is one of the major proteolytic enzymes and is distributed throughout the body. Thus, the assessment of peptide stability toward trypsin degradation is a simple and practical secondary screening assay that can provide valuable information about peptides t_1/2_ and their potential for further development. To investigate the stability of OL-CTOP toward proteolytic degradation, we incubated OL-CTOP with immobilized trypsin, finding an early (20 min) release of CTOP with the additional cyclic fragment f-*cyclo*(CYwOTX)TC(CFR) after a 1 h incubation with trypsin. Importantly, this trypsin degradation fragment contains the active CTOP sequence, and we found its quantity increases over a 16 h incubation to become the major degradation product. Despite the observed proteolytic degradation of the OL part of the OL-CTOP, the cyclic CTOP sequence showed remarkable stability, allowing binding to MOR and antagonist activity. Proteolytic hydrolysis of OL-CTOP and stability of the main proteolytic fragment may play key roles in OL-CTOP in vivo activity. However, further studies are needed to assess proteolytic stability in brains harvested from subjects treated with intranasal OL-CTOP.

Importantly, following intranasal administration, OL-CTOP antagonized the analgesic effects of i.c.v. morphine in mice. Notably, even though a similar dose administered directly in the brain antagonized morphine-induced antinociception, intranasal administration of 600 μg of the parent compound CTOP itself was ineffective, confirming the need for the OL modification to facilitate intranasal administration. Mice treated intranasally with OL-CTOP demonstrated a significant dose-dependent reduction in the antinociceptive effect of morphine. OL-CTOP mediated antagonism was MOR-selective, without effect at KOR or DOR at the dose tested. These findings are consistent with previous reports of CTOP-mediated antagonism [7,8,9], suggesting a similar mode of action for both OL-CTOP and CTOP. Through its actions on MOR expressed on respiratory neurons in the brainstem, morphine depresses ventilation [64]. This respiratory side effect, in some situations, poses severe health risks and limits morphine’s therapeutic use. In our study, mice receiving OL-CTOP pretreatment before morphine showed no significant differences in respiration rate as compared to vehicle-treated mice. Interestingly, OL-CTOP dose-dependently increased respiration rate. Notably, this increase is consistent with reports where MOR antagonists increased respiration through the displacement of endogenous MOR peptides that naturally serve to regulate breathing rate [65], further confirming OL-CTOP mediated MOR antagonism presently.

The reason for the failure to achieve full MOR antagonist activity at higher doses of OL-CTOP is unclear. However, injections of CTOP itself into the ventral tegmental area (VTA) reportedly enhanced extracellular dopamine (DA) levels in the nucleus accumbens (NAc) [66,67]. Given that the activation of postsynaptic DA receptors in NAc neurons reportedly suppresses pain [68,69,70], it is conceivable that OL-CTOP reaching the VTA via transport from the nasal cavity presently produced limited synergistic antinociception, capping the magnitude of antagonism observed. Admittedly, the minimal, brief increase in tail-withdrawal latency induced by OL-CTOP itself in the present study calls this interpretation into question. Further studies of these effects are needed, although they lie outside the scope of this current initial characterization with the limited available OL-CTOP. Likewise, additional tests of OL-CTOP to protect against or reverse opioid-induced respiratory depression or overdose would be of potential interest.

Similar to oxytocin, OL-CTOP could exploit the olfactory and/or trigeminal cranial nerve systems for direct delivery to the brain following intranasal administration. As suggested by the binding study with ASF (Figure 3), OL-CTOP could possibly interact with monosaccharides expressed on olfactory nerves [21] via the odorranalectin part of the molecule, facilitating its intranasal delivery to the brain. Since CTOP peptide alone did not show activity in mice following i.n.s. administration, it is unlikely that OL-CTOP interacts with MOR present in the trigeminal nerve [71,72] system via CTOP sequence and use this system for nose-to-brain transit. In addition, OL-CTOP undergoes proteolytic degradation resulting in a stable fragment that contains the CTOP sequence, suggesting the possibility that OL-CTOP may act as a prodrug. Once internalized into the brain, trypsin-like proteases present in the brain hydrolyze OL-CTOP to release the MOR active CTOP sequence. Studies to elucidate the exact OL-CTOP mode of nose-to-brain transit and action are currently underway.

The reported data clearly demonstrated the feasibility of our total solid-phase synthetic strategy for the preparation of complex cyclic peptides such as OL-CTOP containing two disulfide bonds and revealed the usefulness of odorranalectin scaffold as a delivery platform for the targeted delivery of peptide drugs into the brain.

## 4. Materials and Methods

### 4.1. Chemicals and Reagents

TentaGel XV RAM resin was obtained from Rapp Polymer (Tuebingen, Germany). Fmoc-protected amino acids and coupling reagents (HOBt, HBTU) were purchased from Chem-Impex (Wood Dale, IL, USA) or Novabiochem (Gibbstown, NJ, USA). Kaiser test was purchased from AnaSpec (Fremont, CA, USA). Triisopropyl silane (TIS), anisole and I_2_ were purchased from Sigma-Aldrich (St. Louis, MO, USA) and were of ACS reagent grade (>99.8% purity). Trifluoroacetic acid (TFA) was purchased from Fisher Scientific (Atlanta, GA, USA) and was of ACS reagent grade (>99.8% purity). Tosylsulfonyl phenylalanyl chloromethyl ketone (TPCK) trypsin was purchased from Thermo Fisher Scientific (Grand Island, NY, USA). Asialofetuin (ASF) was purchased from Sigma-Aldrich (St. Louis, MO, USA). HEPES sodium salt was purchased from Fisher Scientific (Atlanta, GA, USA). All solvents and other chemicals were purchased from Fisher Scientific (Atlanta, GA, USA) or Sigma-Aldrich (St. Louis, MO, USA) and were high-performance liquid chromatography (HPLC) grade.

### 4.2. Peptide Synthesis

All linear peptidyl-resin precursors for bicyclic OL-CTOP peptide were synthesized by Fmoc-SPPS on TentaGel XV RAM resin (substitution 0.2 mmol/g, 0.25 mmol scale) using an automated peptide synthesizer (Gyros Protein Technologies PS3 peptide synthesizer, Tucson, AZ, USA). Amino acid couplings were completed by using fourfold excess of amino acids and coupling reagents (HBTU/HOBt) in the presence of 0.4 M NMM in DMF. Fmoc-deprotection cycles were carried out using 20% piperidine in DMF solution. Solid-phase cyclization of linear precursors via disulfide bonds was carried out in a manual reaction vessel. In all cases, Cys/Pen sidechain protecting groups were removed in situ during the final cyclization steps (disulfide bridge formations) with I_2_ (0.5–2 eq), DMSO (0–150 eq) and anisole (4.4 eq) in CH_2_Cl_2_ or DMF (10 mL). All peptides were cleaved from the resin, and all acid-sensitive sidechain-protecting groups were simultaneously removed using TFA/TIS/H_2_O (95:2.5:2.5, *v*/*v*/*v*). Analytical RP-HPLC analyses and peptide purifications were performed on 1260 Infinity (Agilent Technologies, Santa Clara, CA, USA) liquid chromatography systems equipped with a UV/Vis detector. For analytical RP-HPLC analysis, a C18 monomeric column (Grace Vydac, 250 × 4.6 mm, 5 mm, 120 Å), 1 mL/min flow rate, and elution method with a linear gradient of 2 → 100% B over 45 min, where A is 0.1% TFA in H_2_O, and B is 0.08% TFA in CH_3_CN was used. For peptide purification, a preparative C18 monomeric column (Grace Vydac, 250 × 22 mm, 10 mm, 120 Å) was used. The elution method was identical to the analytical method except for the flow rate, which was 15 mL/min. MALDI-TOF mass spectrometry was performed on the Bruker Microflex LT system (Bruker, Billerica, MA, USA) in a reflector mode using an α-cyano-4-hydroxycinnamic acid matrix (positive-ion mode).

### 4.3. Peptide Stability

Proteolytic digestion of OL-CTOP was performed using tosylsulfonyl phenylalanyl chloromethyl ketone (TPCK) trypsin immobilized on agarose resin (Thermo Fisher Scientific, Grand Island, NY, USA), according to the manufacturer’s recommendations. Immobilized trypsin helped eliminate enzyme digest contamination and thus simplify sample analysis. Briefly, 100 μL of resin containing 20 TAME (*p*-toluenesulfonyl-L-arginine methyl ester) units of immobilized TPCK trypsin was washed 3 times with 500 μL of digestion buffer (0.1 M NH_4_HCO_3_ [pH 8.0]); the gel was then suspended in 200 μL of digestion buffer, and 1 mg of OL-CTOP was added for proteolytic digestion. After incubation at 37 °C for 20 min, 1 h, and 16 h, 20 μL of treated samples was collected and analyzed by analytical RP-HPLC and MALDI-TOF MS (Appendix A).

### 4.4. Isothermal Titration Calorimetry (ITC) Experiments

A binding study was performed at 25 °C in 20 mM HEPES at pH 7.0 by using a titration calorimeter PEAQ-ITC (Malvern, Northampton, MA) with a reaction cell volume of 300 μL. Typically, asialofetuin solutions (240 μM) were in the reaction cell and titrated with solutions of OL-CTOP at a concentration of 4000 μM. OL-CTOP and asialofetuin were dialyzed and prepared in the same buffer. At least 18 consecutive injections of 2 μL were applied every 120 s interval at a constant stir speed of 750 rpm. The concentrations of OL-CTOP and ASF were confirmed by measurements using a BioTek Epoch microplate spectrophotometer (Agilent Technologies, Santa Clara, CA, USA). The raw integrated heat plots were analyzed using the MicroCal PEAQ-ITC software v1.21 (Malvern) under the 1-set-of-sites model, and the control parameter as a fitted offset was applied to each titration as per the manufacturer’s guidelines and our previous applications [19,20]. The reported thermodynamic parameters were derived from two independent experiments and then averaged.

### 4.5. Behavioral (In Vivo) Pharmacology

#### 4.5.1. Animals

Experiments were performed on 117 young adult (7–10 weeks old) male C57BL/6J mice obtained from the Jackson Laboratory, Bar Harbor, Maine (USA). All mice were housed 5 per cage in a temperature- and humidity-controlled room at the University of Florida vivarium (Gainesville, FL, USA) on a 12:12-h light/dark cycle with free access to food and water except during experimental sessions. All procedures were preapproved and carried out in accordance with the Institutional Animal Care and Use Committee at the University of Florida as specified by the 2008 National Institutes of Health Guide for the Care and Use of Laboratory Animals. Subjects were assigned to groups randomly, and drug experiments were conducted in a blinded fashion.

#### 4.5.2. Compound Preparation and Administration

All solutions for animal administration were prepared fresh daily. All mice were lightly anesthetized with isoflurane (0.4%) prior to compound administration. Intranasal (i.n.s.) administration was performed with a 20 μL Pipetman, placing a 6 μL drop of OL-CTOP dissolved in sterile saline (0.9%) in each nostril of the inverted animal for a total volume of 12 μL. Intranasal administration took no more than 15 s. Intracerebroventricular (i.c.v.) injections of morphine (10 nmol, in 0.9% sterile saline), U50,488 (100 nmol, in 0.9% sterile saline) or SNC-80 (100 nmol, in DMSO) or CTOP itself (637 μg (3 nmol), in 0.9% sterile saline) were made directly into the lateral ventricle according to the modified method detailed previously [73]. The volume of all i.c.v. injections was 5 μL, using a 10-μL Hamilton microliter syringe. An incision was made in the scalp of the lightly anesthetized mouse, and the injection was made 2 mm lateral and 2 mm caudal to bregma at a depth of 3 mm.

#### 4.5.3. Mouse 55 °C Warm-Water Tail Withdrawal Test

The 55 °C warm-water tail-withdrawal test was performed as previously described [73]. The distal 2.5 cm of a mouse tail was immersed in warm (55 °C) water contained in a 1.5-L heated water bath. The time from the onset of the noxious heat stimulus to the withdrawal of the tail from the heat source was recorded. After determining each mouse’s baseline latency for tail withdrawal, mice were treated with a graded i.n.s. dose of OL-CTOP and/or i.c.v. morphine, U50, 488 or SNC-80. OL-CTOP was administered intranasally (13.5, 45, 135 and 270 nmol, 6 μL each nostril). Each drug-treated group consisted of 5–8 mice, and the tail withdrawal latency for each mouse was measured repeatedly across 45 min. An opioid was then administered to assess the opioid receptor antagonist activity of OL-CTOP. A shortened tail-withdrawal latency after morphine administration indicates antagonist activity of OL-CTOP. Note that control experiments also examine the effect of the parent peptide CTOP itself, without the OL modification, administered either intracerebroventricularly (637 μg) or intranasally (600 μg).

#### 4.5.4. Assessment of Breathing Rate in Mice

Respiration rates (in breaths per minute) were assessed using the Oxymax/CLAMS system (Columbus Instruments, Columbus, OH, USA) as described previously [27]. Mice were habituated to their individual sealed housing chambers for 60 min before testing. Mice were administered morphine (20 mg/kg, i.p.) or vehicle, as indicated, and 5 min later, confined to the CLAMS testing chambers. Additional mice received pretreatment with saline (i.n.s.) or OL-CTOP (100 or 600 μg, i.n.s.) 45 min prior to morphine (20 mg/kg, i.p.). Pressure monitoring within the sealed chambers measured the frequency of respiration. Respiration data were averaged over 20 min periods for 120 min post-i.p. injection of the saline or morphine. Data are presented as % vehicle response ± SEM, breaths per minute.

### 4.6. Statistical Analysis

All tail-withdrawal latency data are reported as mean ± SEM with significance set at *p* < 0.05. Significant differences in behavioral data were analyzed by ANOVA (one-way or two-way with repeated measures as appropriate), with significant results further analyzed with Dunnett’s, Sidak’s or Tukey’s multiple comparison post hoc tests as appropriate using Prism 9.0 software (GraphPad Software, La Jolla, CA, USA).

## 5. Conclusions

In conclusion, we have optimized solid-phase synthesis of a bicyclic peptide OL-CTOP containing two disulfide bridges and demonstrated that this peptide could elicit MOR-antagonism in mice after i.n.s. administration. The differences in reactivity of Cys and Pen thiol groups protected with Trt and Acm protecting groups toward I_2_ in different solvents were exploited for selective disulfide bond formation on the solid phase. The single step and the sequential strategies were used for macrocyclization reactions to generate the desired OL-CTOP. Although both strategies led to the desired OL-CTOP, the sequential strategy yielded a large quantity and better purity of crude bicyclic peptide product. Importantly, intranasally administered OL-CTOP dose-dependently antagonized the analgesic effect of morphine administered to mice through the intracerebroventricular route and prevented morphine-induced respiratory depression.

We believe that the reported synthetic strategies can be applied to the synthesis of a variety of peptides containing multiple disulfide bridges and suitable for intranasal administration, opening new possibilities for the design and discovery of peptide-based drugs targeted to the brain. Additional assessment of the full potential of the OL scaffold as a novel platform for intranasal drug delivery is currently underway in our group.

## 6. Patents

US Patent No. 11,578,100 B2, resulting partially from the work reported in this manuscript, was submitted by J.P.M. and P.C. and published 14 February 2023.

## Data Availability

The datasets generated for this study are available on request to the corresponding authors.

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
