# Peer review of "Solid-Phase Synthesis of the Bicyclic Peptide OL-CTOP Containing Two Disulfide Bridges, and an Assessment of Its In Vivo μ-Opioid Receptor Antagonism after Nasal Administration"

_molecules, 2023, doi:10.3390/molecules28041822_

Round 1
Reviewer 1 Report
In the manuscript, the authors propose the design, synthesis and in vivo evaluation of bifunctional opioid antagonist and odorannalectin mimic for intranasal administration.
Two concepts appeared to be defended in both title and abstract: first, the development of a synthetic strategy allowing access a bicyclic peptide containing 2 disulfide bridges, and second, the design of a fused peptide containing the CTOP sequence for opioid antagonism, inserted into odorranalectin-like peptide sequence for intranasal delivery. If the second is clearly an original and highly interesting approach and molecular concept in mu-opioid targeting, the synthetic strategy development implies chemical tools that are already described both in solution phase and on-resin (references on Albericio’s work/review and recent work of Stockdill – Tetrahedron Letters 2019 - on conotoxins should be added). The novelty on the second point is therefore debatable, even though remarkable yields are obtained and the development is thoroughly discussed.
It would be therefore preferable to reformulate the sentence in line 66-67 “…and to demonstrate that complex cyclic peptides containing multiple disulfide bonds, can be efficiently synthesized using a solid-phase synthetic strategy…”. With the word “demonstrate”, one understands that the authors are providing for the first time, a general procedure for on-resin multi-disulfide bridged peptides. Since the approach was applied to only one sequence, its reproducibility in other sequences is questioned.
Overall, it is a really interesting paper, despite the raise of some unanswered questions regarding the design of the OL-CTOP peptide as potent intranasally administered opioid antagonist and the structure-activity relationship:
- The authors should precise the reason of selecting only the YASPK fragment of odorranalectin in the OL-CTOP for improved intranasal delivery of the CTOP sequence.
- Some random questions: Was there any in vitro data for the OL-CTOP? Have you evaluated the affinity and selectivity of your bicyclic peptide and its metabolites? Have you evaluated the antagonism of CTOP alone? Have you compared the 3D-structure of your peptide with the odorranalectin? -> the answers to these questions are not expected in the manuscript since the authors have chosen to mainly discuss the bicyclic peptide solid-phase synthetic strategy.
Some typo should be corrected :
- Line 21 : trytil -> trityl
- Line 111 : Cys(Acm)6 -> Cys(Acm)6
- Line 380 : “In in all cases…” -> doubling of “in”?
- Line 384: TFA/TIS/H2O -> TFA/TIS/H2O -> number in index
- Incomplete legend in figure S8 in ESI : "Expected [M+H]+= ?? Da, observed..."
Additional remarks:
- Peptides are numbered in figure 1 but are not referred to in the text. Please, insert peptides number in text for better reading.
Author Response
Please find enclosed our revised manuscript entitled “ Solid-phase synthesis of the bicyclic peptide OL-CTOP con-taining two disulfide bridges, and assessment of its in vivo μ-opioid receptor antagonism after nasal administration” by Ramanjaneyulu Rayala, Annika Tiller, Shahayra A. Majumder, Heather M. Stacy, Shainnel O. Eans, Aleksandra Nedovic, Jay P. McLaughlin and Predrag Cudic
We would like to thank you to the reviewers for their constructive and insightful comments, which have helped us to significantly improve our manuscript.
Our responses to the reviewers’ comments are listed below.
Reviewer #1:
We appreciate the reviewer’s comment that this work is a “really interesting paper”.
- References on Albericio’s work/review and recent work of Stockdill-Tetrahedron Letters 2019 - on conotoxins should be added.
Response: The references reflecting work of Albericio and Stockdill are added to the manuscript (references #: 44-48, 53 and 55)
- It would be preferable to reformulate the sentence in line 66-67.
Response: The sentence in line 66-67 is now reformulated and the part of the sentence with the word “demonstrate” is deleted.
- The authors should precise the reasons of selecting only the YASPK fragment of odorranalectin in the OL-CTOP for improved intranasal delivery of the CTOP sequence.
Response: We apologize to the reviewer for lack of clarity on the OL-CTOP design. The YASPK is a part of the OL sequence that remain unmodified. A paragraph was added, lines 306-314 to clarify the OL-CTOP design.
- Was there any in vitro data for the OL-CTOP? Have you evaluated the affinity and selectivity of your bicyclic peptide and its metabolites?
Response: These are very valid questions. We believe that OL-CTOP acts as a prodrug. Based on the literature reports on structures of opioid peptide ligands, OL-CTOP may be too bulky to bind to m-opioid receptor. In order to achieve the desired opioid activity, the active CTOP component needs to be released from OL-CTOP by activity of trypsin-like proteases that are present in brain. Further studies are needed to determine which proteases are involved in OL-CTOP degradation in vivo and the exact structures of OL-CTOP metabolic products. The suggested in vitro studies are planned and will be published elsewhere. Our intention with the submitted manuscript was to report a proof of principle showing that CTOP can be delivered intranasally using OL-based delivery platform, and as such can elicit the desired opioid activity.
- Have you evaluate antagonism of CTOP alone?
Response: While CTOP itself is well characterized (as we reviewed in the manuscript), we agree a direct comparison of the parent peptide is a fair request. In response, we now include a comparison control examining the effects of the parent peptide CTOP alone in vivo, administering a ~600 μg dose through either an intracerebroventricular (i.c.v.) or intranasal (i.n.s.) route. While the direct (i.c.v.) administration of the CTOP to the brain demonstrated the expected morphine antagonism, intranasal administration of CTOP itself proved ineffective (see Figure 3A). These results confirm the need for OL to facilitate transport of the peptide CTOP into brain.
- Have you compared the 3D structure of your peptide and the odorranalectin?
Response: We appreciate this reviewer’s question. Elucidation of the OL-CTOP and odorranalectin 3D structures is important and will be subject of our future studies. Our goal with this manuscript was to demonstrate the usefulness of oddoranalectin as an intranasal delivery platform for CTOP.
- Some typos should be corrected.
Response: We apologize to the reviewer for these unintentional errors. All listed errors are corrected, and the revised manuscript reflects these corrections.
- Peptides are numbered in figure 1 but are not referred to in the text. Please insert peptides numbers in text for better reading.
Response: We apologize to the reviewer for these unintentional errors. The peptide numbers are now referred to in the text, and the revised manuscript reflects these corrections.
Reviewer 2 Report
The paper by R. Rayala et al. describes the design, synthesis and in vivo evaluation of a chimeric peptide containing CTOP, an antagonist of mu opioid receptor.
There are several shortcomings that do not favor publication.
The main one is the design of the hybrid molecule.
First, the rationale is to add to CTOP the lectin-like properties of odorranalectin that should help adsorption to the brain. However, a significant essential part of odorranalectin, the loop segment 9-15, is absent in the hybrid molecule: according to an Ala-scan study by the same Authors (ref. 20), this segment contains residues essential to carbohydrate binding (Tyr9, Gly12, Leu14). Therefore, the odorranalectin portion present is not expected to play the claimed role.
Second, it is claimed that the bicyclisation would better stabilize the compound toward proteolysis. But, in the presence of trypsin, the odorranalectin part is fully and rapidly destroyed: first the extra-cyclic part (which also is not N-blocked, and therefore is susceptible to exopeptidase action) is removed, then the intracyclic “Rf” bond is cleaved.
Third, it is suggested that “incorporation of the CTOP sequence into the odorranalectin scaffold may facilitate potent MOR antagonism. But this has not been checked in vitro (either receptor binding or assays on MVD/GPI).
Fourth, there is no experiment comparing CTOP and OL-CTOP: so not able to demonstrate the interest to design such hybrid molecule.
Concerning the synthesis, the strategy followed for the sequential formation of the two disulfide bonds might not be the most secure. Why not selectively removing the Trt groups followed by disulfide formation ? This would prevent the partial deprotection of Cys(Acm). In fact, concerning the two secondary products 2 and 3, it is not proved that the free Cys (or Pen) is Cys6 or Cys17. And the disulfide bond present in these products might be formed between either Cys6 or Cys17 and either Cys10 or Pen15. And there could be a mixture of mono-diS species.
The Authors assume that the selectivity of diS formation comes from difference in oxidation rates of Cys and Pen. But here the Pen residue pairs with a Cys. So in this case, is this difference determining ?
Other points.
Several sentences/paragraphs should be re-written for better clarity (avoid shortcuts).
The discussion in lines 336-347 is not clear.
Concerning the effect on morphine-induced respiratory depression, it seems that this depression is followed by an over-ventilation after 60-80 min. And this is not reversed by OL-CTOP. What is the explanation for this over-ventilation and the absence of reversal ?
Minor points.
Line 160: trypsin cleaved the peptide between Arg8 and phe9, not Tyr9.
Line 173: the sentence is cut.
Line 256: single step formation of multiple disulfide bonds can be performed using GSH/GSSG or Cystein/Cystin.
Lines 479-480: Figure S3 is not cited.
Line 522: journal name should be abbreviated. And in other references too.
Lines 547-548: reference 20 incomplete.
Caption to Figure S1 says “following Cys6 and Cys17 oxidation” but the structure shows compounds with a disulfide bond between Cys10 and Pen15.
Caption to Figure S1 says “following Cys10 and Pen15 oxidation” (which should be the first oxidation) but the structure is that of the final bicyclic compound.
Author Response
Please find enclosed our revised manuscript entitled “ Solid-phase synthesis of the bicyclic peptide OL-CTOP con-taining two disulfide bridges, and assessment of its in vivo μ-opioid receptor antagonism after nasal administration” by Ramanjaneyulu Rayala, Annika Tiller, Shahayra A. Majumder, Heather M. Stacy, Shainnel O. Eans, Aleksandra Nedovic, Jay P. McLaughlin and Predrag Cudic
We would like to thank you to the reviewers for their constructive and insightful comments, which have helped us to significantly improve our manuscript.
Our responses to the reviewer’s comments are listed below.
Reviewer #2:
- First, the rationale is to add to CTOP the lectin-like properties of odorranalectin that should help adsorption to the brain. However, a significant essential part of odorranalectin, the loop segment 9-15, is absent in the hybrid molecule: according to an Ala-scan study by the same Authors (ref. 20), this segment contains residues essential to carbohydrate binding (Tyr9, Gly12, Leu14). Therefore, the odorranalectin portion present is not expected to play the claimed role.
Response: We apologize to this reviewer for lack of clarity on the OL-CTOP design. Our studies (refs no. 19 and 20) and study of others (ref no. 18) showed that odorranalectin preferentially binds L-fucose and to a lesser extent D‑galactose and N‑acetyl-D-galactosamine, monosaccharides present on the olfactory nerves. NMR binding studies reported in ref 18 showed that residues Lys5, Cys6, Phe7, Cys16 and Thr17 were important for fucose binding. Thus, we hypothesize that replacement of the OL amino acid sequence that is not directly involved in fucose binding with CTOP may provide a novel and unique analogue with desired opioid activity and preserved fucose affinity for successful nose-to-brain delivery. A paragraph was added, lines 306-314, to clarify the OL-CTOP design.
- Second, it is claimed that the bicyclisation would better stabilize the compound toward proteolysis. But, in the presence of trypsin, the odorranalectin part is fully and rapidly destroyed: first the extra-cyclic part (which also is not N-blocked, and therefore is susceptible to exopeptidase action) is removed, then the intracyclic “Rf” bond is cleaved.
Response: Forgive us, but we are surprised with this comment. Nowhere in the text did we state that the bicyclisation would better stabilize the compound toward proteolysis. On the contrary, based on the structures of the know peptide-based opioid ligands, including the CTOP, and the described OL-CTOP stability study toward proteolytic degradation, we believe that OL-CTOP acts as a prodrug. The active CTOP sequence needs to be released from the OL-CTOP molecule in order to achieve the desired opioid activity. However, we share the sentiment for additional studies to determine which proteases are involved in OL-CTOP metabolism in the brain and the structures of the OL-CTOP metabolic products. These studies are underway, but given their understandable complexity, will be published elsewhere.
- Third, it is suggested that “incorporation of the CTOP sequence into the odorranalectin scaffold may facilitate potent MOR antagonism. But this has not been checked in vitro (either receptor binding or assays on MVD/GPI).
Response: in vitro activity of CTOP is well documented in literature (e.g. refs no. 7-9). As mentioned in our response to critique No. 2, we believe that OL-CTOP acts as a prodrug. In other words, the active CTOP sequence needs to be released from the OL-CTOP molecule in order to achieve the desired opioid activity. Absence of the required proteolytic degradation of OL-CTOP, will likely result in no binding to MOR. Since CTOP does not penetrate the brain after systemic administration, and intracerebroventricular (i.c.v.) injection is the only administration route described in the literature, our goal with the reported pilot study was to show that intranasal delivery could be a viable alternative route for CTOP delivery to the brain. In our opinion, the reported in vivo studies provide more useful information on the potentials of OL-CTOP for intranasal delivery and morphine antagonism.
- Fourth, there is no experiment comparing CTOP and OL-CTOP: so not able to demonstrate the interest to design such hybrid molecule.
Response: Although it is well established that the parent peptide CTOP has exceptionally poor CNS penetrant abilities alone (as reviewed), we agree that as direct comparison of CTOP and OL-CTOP would strengthen the benefits of the current analog. In response, we now include a comparison control examining the effects of the parent peptide CTOP alone in vivo, administering the 600 μg dose through an intranasal (i.n.s.) route. While the i.n.s. administration of OL-CTOP antagonized morphine administered to the brain, intranasal administration of the parent peptide CTOP itself proved ineffective (see Figure 3A). These results confirm the need for OL to facilitate transport of the peptide CTOP into brain, and (we believe) provide the requested rationale for the work. These comparisons are now included in the results and discussion.
- Concerning the synthesis, the strategy followed for the sequential formation of the two disulfide bonds might not be the most secure. Why not selectively removing the Trt groups followed by disulfide formation? This would prevent the partial deprotection of Cys(Acm). In fact, concerning the two secondary products 2 and 3, it is not proved that the free Cys (or Pen) is Cys6 or Cys17. And the disulfide bond present in these products might be formed between either Cys6 or Cys17 and either Cys10 or Pen15. And there could be a mixture of mono-diS species.
Response: Our goal was to demonstrate that the bicyclic OL-CTOP peptide can be synthesized using solid-phase strategy, including solid-phase Cys/Pen oxidation with I2, which is one of the most affordable reagents for this purpose. However, to minimize Acm removal and formation of other side products, the oxidation reaction had to be optimized by adjusting the I2 molar equivalents. Since I2 is also used in peptide chemistry as a reagent for Acm removal, selective removal of Trt groups follow by disulfide formation will not prevent partial Acm removal from the peptidyl resin precursor. As shown in Figure 1B, HPLC traces of a crude peptide show formation of only one major product after oxidation with I2. Formation of the one major bicyclic peptide, and not a mixture of bicyclic products, can be explained by large selectivity (cca 100 fold) between S-Trt and S-Acm in CH2Cl2 and DMF (ref 54).
- The Authors assume that the selectivity of diS formation comes from difference in oxidation rates of Cys and Pen. But here the Pen residue pairs with a Cys. So in this case, is this difference determining?
Response: We believe that selectivity comes from differences in oxidation rates between S-Trt and S-Acm in CH2Cl2 and DMF (ref 54), not Cys and Pen. Considering the oxidation reaction mechanism with I2, we believe that the structural differences between Cys and Pen do not significantly affect the oxidation reaction rate.
- Other points. Several sentences/paragraphs should be re-written for better clarity (avoid shortcuts). The discussion in lines 336-347 is not clear.
Response: We apologize to the reviewer for lack of clarity in this paragraph. The paragraph is now revised for clarity and the current text reflects this change.
- Concerning the effect on morphine-induced respiratory depression, it seems that this depression is followed by an over-ventilation after 60-80 min. And this is not reversed by OL-CTOP. What is the explanation for this over-ventilation and the absence of reversal?
Response: This is an astute remark; thanks. Responding in segments: “rebound” of respiration is a typical effect for organisms experiencing morphine-induced respiratory depression; this has been observed repeatedly in our previous tests (for instance, see Brice-Tutt et al., Br J Pharmacol, 2020) and those of others. Moreover, the over-ventilation induced by OL-CTOP is characteristic of treatment with MOR antagonists, and has long been attributed to the displacement of endogenous opioid peptide agonists working through the mu opioid receptor to naturally suppress respiration (see Isom and Elshowihy, 1982 for an early demonstration of this). We now add additional testing with a higher (600 μg, i.n.s.) dose of OL-CTOP to confirm this effect, and now include discussion of this observation in the discussion section.
- Minor points.
Line 160: trypsin cleaved the peptide between Arg8 and phe9, not Tyr9.
Line 173: the sentence is cut.
Line 256: single step formation of multiple disulfide bonds can be performed using GSH/GSSG or Cystein/Cystin.
Lines 479-480: Figure S3 is not cited.
Line 522: journal name should be abbreviated. And in other references too.
Lines 547-548: reference 20 incomplete.
Caption to Figure S1 says “following Cys6 and Cys17 oxidation” but the structure shows compounds with a disulfide bond between Cys10 and Pen15.
Caption to Figure S1 says “following Cys10 and Pen15 oxidation” (which should be the first oxidation) but the structure is that of the final bicyclic compound.
Response: We apologize to the reviewer for these unintentional errors. All listed errors are corrected, and the revised manuscript reflects these corrections.
Round 2
Reviewer 2 Report
Overall, whatever the in vivo results, I think that it is important to support the rational design and the hypothesized mechanism of action by more experimental data.
I have the following comments to some of the Authors’ answers.
Comments to response point 1:
As a significant part of odorranalectin was absent in the chimeric peptide, it is of course important to consolidate the design rational and to clearly mention the arguments provided by the Authors’ response (should be placed in the Introduction or at the beginning of the Results section). The fact is that it is said in the abstract of reference 20 that “Results revealed that Lys5, Phe7, Tyr9, Gly12, Leu14, and Thr17 were crucial for binding BSA-Lfucose”. And the Ala-scan might be more precise than the NMR binding studies as the later shows direct interactions only, but the same study also indicated H-bonds that might be important for peptide conformational stability. Also, unfortunately, there is no experiment showing that the OL-CTOP peptide indeed binds to fucose.
Comments to response point 2:
At the beginning of the Discussion, there is a long paragraph about the benefits of peptide cyclisation to increase biological activity through stabilization of its conformation and toward degradation by peptidases. In addition, the Authors insisted about disulfide-rich peptides, which are attractive targets for drug discovery. So, it looked logical to think that the Authors designed such bicyclic peptide for better properties.
I am also surprised by the Authors’ response because nowhere was mentioned the possibility that OL-CTOP could act as a prodrug. Nowhere, it is said that “The active CTOP sequence needs to be released from the OL-CTOP molecule in order to achieve the desired opioid activity.”And again surprisingly, this is not more discussed in the new version.
Also it is said that “Thus, assessment of peptide stability toward trypsin degradation is a simple and practical secondary screening assay that can provide valuable information about peptides t1/2 and their potentials for further development.”, suggesting that the Authors would have prefered stable compound.
Comments to response point 3:
It seems that the Authors hypothesize that OL-CTOP would not be able to bind to MOR and again that OL-CTOP would be just a pro-drug releasing CTOP after penetration and digestion. This was not mentionned in the first version, it is not also the case in the new one. Again, only an experiment could clearly show the MOR binding properties of the chimeric peptide.
Comments to response point 4:
That OL helps intranasal delivery seems indeed the case. But that this is due to binding to sugars including fucose remains to be demonstrated: as already said, in vitro binding to fucose should be checked. And replacing the OL part by sequence that do not bind sugars (eg replacing a crucial residue by an Ala).
Author Response
Response to Reviewers Comments
Reviewer 1:
Reviewer 1 was satisfied with previous edits, and had no further comments.
Reviewer 2:
Overall, whatever the in vivo results, I think that it is important to support the rational design and the hypothesized mechanism of action by more experimental data.
…While we are gratified the reviewer supports the value of the in vivo studies, we now incorporate more description of the philosophy guiding the design of the OL-CTOP (see points below). Notably, the addition of new experimental ITC data now provides the experimental data requested.
1) As a significant part of odorranalectin was absent in the chimeric peptide, it is of course important to consolidate the design rational and to clearly mention the arguments provided by the Authors’ response (should be placed in the Introduction or at the beginning of the Results section). The fact is that it is said in the abstract of reference 20 that “Results revealed that Lys5, Phe7, Tyr9, Gly12, Leu14, and Thr17 were crucial for binding BSA-Lfucose”. And the Ala-scan might be more precise than the NMR binding studies as the later shows direct interactions only, but the same study also indicated H-bonds that might be important for peptide conformational stability.
… We agree with this reviewer’s comment that intramolecular H-bonds are important for peptide conformational stability, as in the case for OL. However, the bicyclic peptide OL-CTOP is structurally different from the cyclic OL and different H-bonds (influenced by the presence of two disulfide bonds) may contribute to its conformational stability. We have now included the isothermal titration calorimetry (ITC) binding study with OL-CTOP and asialofetuin (ASF), a model glycoprotein that possesses the same terminal sugar residues as the olfactory epithelium, and demonstrated that OL-CTOP binds ASF with affinity comparable to OL (within the limits of experimental error). In addition, we showed that OL-CTOP/ASF interaction is an enthalpy-driven process, which is typical for interactions between lectins and carbohydrate ligands as previously described in literature (Refs 29 and 30).
2) Also, unfortunately, there is no experiment showing that the OL-CTOP peptide indeed binds to fucose.
…While we have shown previously that odorranalectin binds to glycoproteins containing different sugar moieties (see ref. 19 and 20 in the manuscript), but acknowledge the reviewer’s concern that OL-CTOP may differ. Responding directly to the reviewer’s desire for binding data, we now also include the results of ITC titration to demonstrate OL-CTOP binding to asialofetuin (ASF), as this glycoprotein is well characterized and possesses the same terminal sugar residues (Gal and GalNAc) as the olfactory epithelium. As demonstrated in Figure 3 (and associated text in the results section), OL-CTOP was found to bind ASF with a Kd value of 176 μM, confirming the glycoprotein interaction as requested.
3) At the beginning of the Discussion, there is a long paragraph about the benefits of peptide cyclisation to increase biological activity through stabilization of its conformation and toward degradation by peptidases. In addition, the Authors insisted about disulfide-rich peptides, which are attractive targets for drug discovery. So, it looked logical to think that the Authors designed such bicyclic peptide for better properties.
…While cyclization is a well-established approach to improve the druggability of linear peptides with otherwise unfavorable PK properties, we acknowledge the reviewer’s point that this body of text may go beyond the focus of the present manuscript. In addition to increased peptide stability, cyclization of peptides introduces conformational constraints that significantly decrease the conformational freedom that may lead to preorganized conformations better suited for binding a target protein (lines 308-311). Accordingly, we have now simplified the opening of the discussion, eliminating the comment on disulfide-rich structures effects on chemical, enzymatic or thermal degradation and focus on the benefits of the cyclic structure for peptides pharmaceutical properties.
4) I am also surprised by the Authors’ response because nowhere was mentioned the possibility that OL-CTOP could act as a prodrug. Nowhere, it is said that “The active CTOP sequence needs to be released from the OL-CTOP molecule in order to achieve the desired opioid activity.” And again surprisingly, this is not more discussed in the new version.
…We did not assess CTOP release in vivo from OL-CTOP, as extensive (and expensive) pharmacokinetic protein studies were deemed to be beyond the scope of this study, where the focus was on detailing the initial synthesis OL-CTOP and demonstration of biological activity. Accordingly, we sought to interpret our data narrowly as befits an initial report of this type. Still, we share with the reviewer a desire for this full characterization and pharmaceutical development of OL-CTOP, and apologize for not responding to this concern in the initial review. We now address reviewer 2’s interpretation in a review of possible CTOP release from OL-CTOP in the discussion section (lines 405-408), noting that this testing would be appropriate for future studies.
5) Also it is said that “Thus, assessment of peptide stability toward trypsin degradation is a simple and practical secondary screening assay that can provide valuable information about peptides t1/2 and their potentials for further development.”, suggesting that the Authors would have prefered stable compound.
…To be clear: we included preliminary proteolytic stability data to address potential concerns over peptide stability in a biological system, demonstrating that the CTOP fragment possesses sufficient half-life to match the biological therapeutic activity demonstrated. On the other hand, proteolytic hydrolysis of the OL part of the OL-CTOP molecule is desirable to release the active CTOP sequence. We believe the cited text as currently presented in the discussion appropriately represents this position.
6) It seems that the Authors hypothesize that OL-CTOP would not be able to bind to MOR and again that OL-CTOP would be just a pro-drug releasing CTOP after penetration and digestion. This was not mentioned in the first version, it is not also the case in the new one. Again, only an experiment could clearly show the MOR binding properties of the chimeric peptide.
…As evident upon review of the introduction (see the last paragraph), we did not hypothesize in any way that OL-CTOP acted as a prodrug or any other such mechanism of action. While we agree this is an interesting question, respectfully, we feel it beyond the scope of this initial report, and a topic better addressed by future studies focusing on this matter. Of course, as made clear in the present manuscript, we also present strong behavioral pharmacology data demonstrating OL-CTOP mediated antagonism of the MOR in mice.
7) That OL helps intranasal delivery seems indeed the case. But that this is due to binding to sugars including fucose remains to be demonstrated: as already said, in vitro binding to fucose should be checked. And replacing the OL part by sequence that do not bind sugars (eg replacing a crucial residue by an Ala).
…We appreciate the reviewer’s acknowledgement that OL facilitates intranasal delivery of CTOP. Addressing the reviewer’s comment, as noted above, we have now included ITC data demonstrating the interaction between asialofetuin (ASF) as a model glycoprotein and OL-CTOP (lines 177-198). ASF possesses Gal and GalNAc terminal sugar residues available for OL and OL-CTOP binding. In addition to Fuc, Gal and GalNAc are also widely distributed on the olfactory epithelium of nasal mucosa. The obtained Kd for OL-CTOP/ASF interaction is comparable (within the limits of experimental error) to Kd previously reported for OL and the thermodynamic parameters are typical for interactions between lectins and carbohydrate ligands (Refs 29 and 30). Notably (again), this data extends our earlier demonstration of this interaction, where the critical odorranalectin residues for binding were mapped with alanine substitution as suggested by the reviewer (see ref 20, Singh et al., Eur J Org Chem., 28 (2022) e202200302). In addition to including the ITC data, we have now emphasized this earlier study of odorranalectin binding to glycans as requested in the introduction section (lines 72-74) and discussion section (lines 177-198 and 340-350) of the current manuscript.
We again thank you for consideration of our work and look forward to your response to this revised original manuscript.
Sincerely yours,
Predrag Cudic, Ph.D.
Professor
Department of Chemistry and Biochemistry
Charles E. Schmidt College of Science
Florida Atlantic University
5353 Parkside Drive, Jupiter, FL 33458